# Position: LLM Agents Are the Antidote to Walled Gardens

**Samuele Marro** [1 2]   **Philip Torr** [1 2]

## Abstract

While the Internet's core infrastructure was designed to be open and universal, today's application layer is dominated by closed, proprietary platforms. Open and interoperable APIs require significant investment, and market leaders have little incentive to enable data exchange that could erode their user lock-in. We argue that LLM-based agents fundamentally disrupt this status quo. Agents can automatically translate between data formats and interact with interfaces designed for humans: this makes interoperability dramatically cheaper and effectively unavoidable. We name this shift *universal interoperability*: the ability for any two digital services to exchange data seamlessly using AI-mediated adapters. Universal interoperability undermines monopolistic behaviours and promotes data portability. However, it can also lead to new security risks, technical debt, and legal frictions. Our position is that the ML community should embrace this development while building the appropriate frameworks to mitigate the downsides. By acting now, we can harness AI to restore user freedom and competitive markets without sacrificing security.

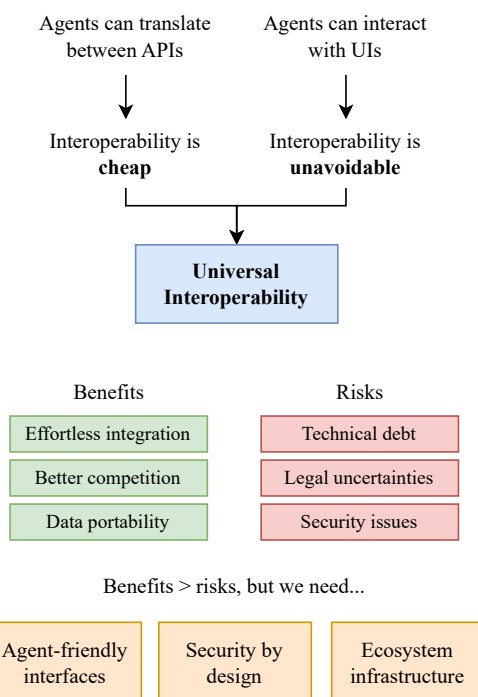

*Figure 1.* Summary of universal interoperability and of our position.

## 1. Introduction

The Internet's core infrastructure (TCP/IP, HTTP, DNS...) was designed to be open and universal (Fall & Stevens, 2012; Gourley & Totty, 2002; Zeldman, 2003). In theory, any two machines anywhere should be able to exchange data and services freely; in practice, the application layer looks very different: disconnected social networks, enterprise software with proprietary APIs, and mobile platforms that restrict developers to closed ecosystems (Frieden, 2016; National Telecommunications and Information Administra-

tion, 2023). While the underlying infrastructure remains largely open, most user-facing services today are effectively **walled gardens** (Staab, 2024).

Part of this fragmentation is purely technical. Building and maintaining robust integrations is time-consuming and expensive (Knoche & Hasselbring, 2021). Every new service or API version requires updates to client libraries, compatibility testing, and ongoing coordination. Integration challenges include handling differing data models (John, 2025), incorporating business rules (Corradini et al., 2018), managing errors (Sheriffdeen & Heart, 2020), and maintaining version compatibility (Lercher et al., 2024). As a result, many potential integrations never materialise simply because the development and maintenance costs outweigh the perceived business benefits.

However, beyond technical barriers, there is a **strategic incentive** to stay closed. Platforms with strong network

[1]Department of Engineering Science, University of Oxford [2]Institute for Decentralized AI. Correspondence to: Samuele Marro <samuele.marro@eng.ox.ac.uk>.

*Proceedings of the 43rd International Conference on Machine Learning*, Seoul, South Korea. PMLR 306, 2026. Copyright 2026 by the author(s).

effects, where the value for each user grows as more people join, can retain their user base by preventing easy data or feature portability (OECD, 2021). Each additional user or developer increases the value of the platform, making it harder for rivals to compete. Over time, the resulting *lock-in* raises switching costs, concentrates power, and reduces competitive pressure on incumbents (Crémer & Biglaiser, 2012).

The third barrier is legal: the terms of service routinely prohibit automated access, and some forms of agent-mediated interoperability operate in a legal grey area (Fiesler et al., 2020; Atkinson, 2025). Recognizing this gap, regulators have begun to intervene. The European Union's Digital Markets Act, for example, requires large messaging services to interoperate with third parties (Parliament, 2022), while GDPR gives users rights to export their personal data in machine-readable formats (European Parliament and Council of the European Union, 2016). While these measures represent positive steps, they tend to be *reactive and narrow*, focused on specific sectors or use cases, and moving slowly through legislative processes (Afina et al., 2024). By the time a new rule takes effect, dominant platforms may have entrenched their positions even further.

At the same time, we are witnessing a turning point driven by Large Language Model (LLM) based agents. First, LLMs have become remarkably adept at interfacing with machine-readable services, such as REST APIs (Song et al., 2023) and SQL databases (Mohammadjafari et al., 2024), tasks that previously required extensive custom code. Second, web-interfacing agents can browse and interact with websites just as a human user would (Zhang et al., 2024a). These agents can perform complex sequences such as clicking buttons, filling in forms, and extracting content, even when a formal API is not present. Unlike traditional web scrapers, they can adapt to changes in UI elements and website structures, making them reliable integration tools.

Together, these two capabilities make interoperability both **much cheaper and effectively unavoidable**. A platform that chooses not to open its API can still be accessed by an agent that interacts with its web interface. This represents a fundamental shift in the balance of power between platforms and users, as agents undermine the traditional gatekeeping role of the former.

We argue that the Machine Learning community should embrace this development, as it has the potential to remove platform lock-in and restore user freedom. However, without proper guidance, this new wave of ad-hoc AI-driven interoperability could create different problems, such as security vulnerabilities, inconsistent implementations, and

new forms of lock-in around specific agent frameworks. Unauthorised data access, unintentional breaches of terms of service, and unpredictable behaviours could undermine trust in the entire ecosystem, while the lack of standards could lead to technical debt and incompatibility. To avoid these issues and unlock the full potential of interoperability, we need a principled foundation. In short (Figure 1):

---

**Position Statement**

**The widespread adoption of LLM agents has the potential to make interoperability affordable and inevitable. We should embrace this shift while developing infrastructure that prevents new technical debt, security risks, and legal frictions.**

---

In this paper, we review the economics of interoperability and the shortcomings of previous interoperability efforts. We then explain how LLM agents disrupt the current landscape and the potential risks and benefits. Finally, we outline how the ML community can lay the groundwork to minimise the issues of universal interoperability, as well as how alternative views can inform this effort.

By establishing these foundations now, we can use agents' capabilities to reduce lock-in and foster a more open, competitive digital environment, effectively "killing" traditional network effects while maintaining security, privacy, and user choice.

## 2. Background: Interoperability

Interoperability, the ability of different systems to exchange and use information effectively, is a cornerstone of computer science. At its most basic level, *syntactic interoperability* requires agreement on data formats and communication protocols (for example, message framing and byte ordering), while *semantic interoperability* requires consistent interpretation of exchanged data (IEEE, 1991; International Organization for Standardization and International Electrotechnical Commission, 2015).

Foundational work in the 1980s, such as the OSI reference model and the POSIX standard, introduced layered abstractions and vendor-neutral interfaces that anticipated modern multi-vendor environments (IEEE, 1996; Zimmermann, 1980). In the early 2000s, web services frameworks brought interoperability to enterprise applications. The SOAP/WSDL stack defined a comprehensive XML-based approach to publishing and invoking remote services, supplemented by WS-Security, WS-ReliableMessaging, and related standards to ensure secure and reliable interactions (World Wide Web Consortium, 2003). Its complexity eventually led to the adoption of the simpler *RESTful* style, which

uses standard HTTP methods (GET, POST, PUT, DELETE) and lightweight payloads (typically JSON) to reduce development overhead (Fielding, 2000). The simplicity of REST led to widespread adoption by major platforms (Amazon, Google, Facebook), while the OpenAPI specification emerged to provide machine-readable interface contracts (OpenAPI Initiative, 2024). At the same time, OAuth 2.0 and OpenID Connect established protocols for delegated authorisation and single-sign-on, which enabled third-party applications to access user data without credential sharing and thus fostered interoperability in web and mobile services (Hardt, 2012; Sakimura et al., 2014). Similarly, the W3C's Semantic Web stack (including RDF (Klyne & Carroll, 2004), OWL (Motik et al., 2008), and SPARQL (Prud'hommeaux & Seaborne, 2008)) and initiatives such as schema.org have enabled ontology-based interoperability by embedding structured metadata in web content (Cyganiak et al., 2014).

## 2.1. Economics of Interoperability

From an economic point of view, interoperability is closely related to *compatibility* between products or networks, a concept explored in the classic literature on industrial organisation. Katz and Shapiro (Katz & Shapiro, 1985) and Farrell and Saloner (Farrell & Saloner, 1992) showed that compatibility decisions influence market outcomes by affecting the switching costs of consumers and the incentives of firms to invest. In markets exhibiting positive network effects, that is, where the value of a product to each user increases with the number of users, firms can restrict interoperability to preserve user lock-in and increase the entry barriers of competitors (Farrell & Saloner, 1992; Katz & Shapiro, 1985). Interoperability mitigates these effects by allowing new entrants to leverage the networks of incumbents, thereby reducing effective switching costs and allowing multi-homing (the ability to use multiple services in parallel) (Crémer et al., 2019). Empirical studies of platform markets confirm that lower switching costs correlate with higher user turnover rates and more vigorous competition (Cheng, 2021), although some researchers challenge the universality of this assumption (Dubé et al., 2009).

A closely related concept is *data portability* (Graef et al., 2018), which reduces the cost of migrating data (e.g., profiles, contacts, content) between platforms. Network-level interoperability and data portability create a powerful dynamic: users face lower technical barriers to communication and fewer logistical challenges to switching services (Engels, 2016; OECD, 2021).

Together, these economic insights indicate that while interoperability is primarily a technical property, it operates as a

strategic mechanism that can reshape market structure, influence investment incentives, and improve consumer welfare in networked industries (Bourreau et al., 2022). Conversely, limited or absent interoperability benefits incumbents by maintaining high switching costs and controlling access to complementary services, thus preserving their user base and limiting competition (OECD, 2021).

## 2.2. Interoperability Efforts

Since 2000, standards bodies and industry consortia have developed technical specifications to address various aspects of interoperability. For example, federated protocols (e.g., XMPP for instant messaging (Saint-Andre, 2011) and ActivityPub for social networking (Lemmer-Webber et al., 2018)) show how open specifications can support decentralised ecosystems, while domain-specific frameworks, such as HL7 FHIR in healthcare (Health Level Seven International, 2025) and ISO 20022 in financial messaging (International Organization for Standardization, 2013), illustrate how targeted standards enable interoperability in regulated sectors.

Recognising the pro-competitive potential of interoperability, regulators have increasingly adopted measures to promote interconnection. In 2004, the European Commission required Microsoft to disclose proprietary protocol specifications under antitrust law, establishing legal precedent for mandated interoperability (European Commission, 2004). The EU General Data Protection Regulation introduced the right to data portability, which requires personal data to be provided in "structured, commonly used, machine-readable" formats (European Parliament and Council of the European Union, 2016). More recently, the EU Digital Markets Act requires "gatekeepers" to interoperate with smaller messaging services on core functionalities (Parliament, 2022). Additional initiatives such as the US ACCESS Act (U.S. House of Representatives, 2021), the 21st Century Cures Act's anti-information-blocking rules (U.S. Congress, 2016) in the US, and the Open Banking Standard in the UK (Open Banking Working Group, 2016), reflect a global trend towards mandated interoperability.

Despite this progress, challenges remain. Technical standards often struggle to achieve full semantic compatibility (Nagarajan et al., 2006), partial interoperability can produce unexpected competitive dynamics (Bourreau et al., 2022), and regulatory interventions often lag behind platform evolution (Afina et al., 2024). As AI-driven agents capable of automating integration at scale emerge, it is essential to build on these interdisciplinary lessons to develop infrastructure that ensures that interoperability remains robust, secure, and effective.

# 3. A Universal Adapter

Recent advances in LLMs have enabled a new class of autonomous systems, namely LLM-based agents (Schick et al., 2023), that combine natural language understanding, code generation, and tool use to bridge heterogeneous interfaces. Due to the multitude of definitions of LLM agents in the literature (Xi et al., 2025), we clarify that in this paper we define an LLM-based agent as any system that (a) understands and produces text in natural language, code, or structured formats (e.g., JSON, SQL) and (b) interacts with external tools or web pages via API calls or simulated user actions. Unlike classical middleware or custom adapters, these agents leverage the broad knowledge and flexible reasoning of LLMs to achieve two key capabilities that undermine traditional barriers to interoperability: **automated translation between formats** and **robust interaction with user interfaces**. In this section, we show how these two capabilities upend the current status quo of interoperability.

## 3.1. Automated Translation Between Formats

LLM agents can automatically translate between diverse data schemas or API specifications with minimal human input. Given two schemas, an LLM agent infers correspondences between fields, even with implied (Sheetrit et al., 2024). Other agents can also emit executable "glue" code to perform the conversion: for example, RestGPT parses REST endpoint descriptions and generates Python code to invoke the service and transform the response into the target format (Song et al., 2023). Similarly, Gorilla retrieves up-to-date API documentation and produces calls for hundreds of APIs, maintaining accuracy even when specifications change (Patil et al., 2024). Beyond REST, LLMs have demonstrated the ability to map from natural language to SQL queries on complex benchmarks (Hong et al., 2024; Gao et al., 2023). Recent work on GraphQL generation further shows that LLMs can generate valid GraphQL queries that satisfy intricate type constraints (Saha et al., 2024). These translation capabilities reduce traditional barriers to interoperability: writing and maintaining client libraries, managing version compatibility, and encoding business rules become single-prompt or few-shot tasks (see (Lehmann, 2024) for an early proposal of LLM-mediated interoperability). Furthermore, once generated, adapter code can be reused programmatically, making this approach highly scalable.

> ### Key Observation #1
>
> LLM agents significantly reduce the technical effort required to integrate an API.

This capability has far-reaching economic implications. By making API interfacing and format translation nearly free, LLM agents eliminate the high engineering costs that traditionally discouraged integration. When translation no longer depends on custom adapters, proprietary APIs lose their strategic advantage: new entrants can easily connect to dominant services, and users can migrate or multi-home without prohibitive development overhead.

## 3.2. Robust Interaction with User Interfaces

Even without formal APIs, LLM agents can interact with web interfaces by reading, analysing, and manipulating the underlying DOM or GUI elements. Previous approaches to web automation, such as XPath-based crawlers (Mirtaheri et al., 2014) or scripted robotic process automation tools (RPAs) (Chakraborti et al., 2020), are brittle and require significant customisation. In contrast, LLM agents guided by frameworks like ReAct combine reasoning and acting: the model inspects page elements, generates dynamic plans, and executes them (Yao et al., 2023). On benchmarks such as MiniWoB (Shi et al., 2017) and WebArena (Zhou et al., 2024), ReAct agents achieve substantially higher success rates on navigation tasks than imitation-based and reinforcement-learning agents.

These UI-capable agents, combined with improving performance on CAPTCHAs (Plesner et al., 2024), ensure that human-facing interfaces cannot effectively prevent machine access. While UI automation is less efficient than direct API calls, it provides sufficient scalability for programmatic control across hundreds of web endpoints with minimal additional effort. As a consequence, deliberately closing platforms through interface obfuscation yields little benefit:

> ### Key Observation #2
>
> Any computer interaction accessible to a human can be replicated by a sufficiently advanced LLM agent.

This observation is supported by empirical evidence. In three years, performance on WebArena rose almost by an order of magnitude, from 8.87% in March 2023 (Zhou et al., 2024) to 71.6% in January 2026 (Guo et al., 2026), and has almost reached the human score of 78.24%. In parallel, every major frontier lab has released a production web agent or an agent browser in the last 18 months, including ChatGPT Atlas (OpenAI, 2025), Claude in Chrome (Anthropic, 2025), Gemini Computer Use (Google DeepMind, 2025), Perplexity Comet (Perplexity, 2025), Copilot Mode in Edge (Microsoft Edge Team, 2025), and Amazon Nova Act (Amazon AGI, 2025).

Together, these advances mean that interactions available

through GUIs or APIs can, in principle, be automated by an LLM agent. Whether a service exposes a machine-readable API or only a web interface, agents render proprietary barriers ineffective at scale.

# 4. Universal Interoperability

Having outlined the traditional interoperability challenges in Section 2 and shown in Section 3 how LLM agents can overcome these barriers, we now formalise *universal interoperability*, i.e., the explicit use of LLM agents for large-scale interoperability:

> ### Universal Interoperability
>
> Universal interoperability is the capability for any two digital services, whether they expose machine-readable APIs or only human-facing interfaces, to exchange data, invoke functionality, and coordinate workflows seamlessly via AI-mediated adapters.

In this framework, an LLM-based "universal adapter" dynamically discovers available operations, infers schema mappings, and generates the necessary glue code or UI actions at runtime, eliminating dependence on pre-programmed integrations.

Universal interoperability differs fundamentally from previous integration paradigms through its use of dynamic, AI-driven adapters rather than static middleware or ontologies. Traditional semantic Web approaches such as RDF and OWL require extensive ontology engineering and schema registries to align disparate data models (Akanbi & Masinde, 2018; Castano et al., 2006). Standards-based API federation (such as WS-I Basic Profile, OpenAPI, and GraphQL) reduces boilerplate but still requires predefined interface contracts. RPA and rule-based crawlers automate GUIs but are vulnerable to layout changes and lack semantic understanding (Chakraborti et al., 2020; Mirtaheri et al., 2014; Weninger et al., 2016). In contrast, universal interoperability uses LLMs at runtime to infer schema correspondences, generate necessary code or UI actions on demand, and adapt to evolving interfaces, transforming weeks of engineering effort into a handful of prompts. This approach reduces maintenance burden and extends seamless integration to any service reachable via API or GUI.

## 4.1. Benefits

The primary advantage of universal interoperability is that it drastically reduces the engineering effort and cost of integrating heterogeneous systems. Traditional data integration often requires extensive schema mapping, custom client li-

braries, and compatibility testing, resulting in a substantial investment of time and money. As discussed in Section 3, LLM-based agents can automatically infer correspondences between specifications and generate glue code, minimising manual effort and reducing maintenance and debugging overhead. Organisations can thus achieve interoperability among each other even when using legacy systems. This is particularly useful for entities with limited IT resources, such as non-profits and local administrations. Furthermore, by reducing switching costs and enabling smooth data portability, universal interoperability fosters more competitive markets and empowers users. As analysed in Section 2.1, interoperability and portability weaken lock-in, increase competition, and improve consumer outcomes. When AI agents bridge two services, users can multi-home without losing data or contacts, and businesses can take advantage of previously unsupported integrations.

Beyond the core interoperability aspects we discussed, universal interoperability offers additional benefits. Lower connection barriers accelerate innovation and democratise integration capabilities: end-user programming research has shown that domain experts like scientists, analysts, and healthcare professionals already create their own automation using visual or rule-based tools (Ur et al., 2016). LLMs now enable these professionals to specify complex tasks using natural language, automatically generating the necessary API calls or scripts.

## 4.2. Risks

However, automating integration through LLM-based agents introduces numerous security risks (Chiang et al., 2025), which are amplified by reduced human oversight. As Bainbridge observed in her "Ironies of Automation," when humans shift from active operators to passive monitors, their ability to intervene during unexpected failures decreases (Bainbridge, 1983). LLM agents exacerbate this by operating autonomously on critical data flows, often without transparent logging or clear escalation mechanisms (South et al., 2025). Furthermore, agents are vulnerable to adversarial attacks: attackers can serve malicious web pages that trick the agent into disclosing credentials or executing unintended actions (Evtimov et al., 2025; Xu et al., 2024; Zhang et al., 2024b). These attack vectors can lead to unauthorized data access, privilege escalation, and undetected system compromise.

In addition, the legal and commercial environment around AI-mediated interoperability remains ambiguous, with platform providers already implementing defensive measures. Many terms of service explicitly prohibit automated scraping (Fiesler et al., 2020), though enforcement remains in-

consistent due to legal uncertainties (Atkinson, 2025). Regardless of the legality of scraping, web providers have deployed anti-bot defences: notably, AI Labyrinth, developed by Cloudflare, redirects suspected AI crawlers through decoy pages containing nonsensical content, depleting attacker resources (Tatoris et al., 2025). As unchecked AI automation proliferates, web services may implement more severe countermeasures, such as data poisoning (Carlini et al., 2024) or increasingly inaccessible CAPTCHA systems (Searles et al., 2023; Tariq et al., 2023).

Even in non-adversarial settings, LLM agents can be affected by small variations in prompts, interfaces, or responses (Loya et al., 2023; Razavi et al., 2025). Without proper safeguards, agents can hallucinate data mappings or omit critical fields (Béchard & Ayala, 2024), leading to silent data corruption. Schema drift and evolving web interfaces further destabilise agent workflows: a minor change in HTML elements or API response formats could break an integration in opaque ways (Zhang et al., 2025). More broadly, LLM agents might generate new forms of technical debt: every prompt template, parsing rule, and auxiliary tool becomes part of the codebase. When models or downstream services are updated, these interdependencies can trigger cascading failures, requiring significant engineering effort to diagnose and solve.

Finally, universal interoperability risks reintroducing lock-in at the agent layer. As an agent infrastructure achieves sufficient scale, it achieves negotiating leverage with service providers, potentially concentrating market power (Agranat & Gal, 2025). Additionally, proprietary agent implementations or closed-source LLMs may favour services with commercial arrangements, reinforcing network effects that disadvantage smaller providers (Kapoor et al., 2025). Thus, rather than eliminating vendor lock-in, AI-mediated interoperability may simply shift it from the API level to the model and agent-framework level.

### 4.3. Is Universal Interoperability Worth It?

After assessing benefits and risks, we conclude that universal interoperability represents an opportunity worth pursuing. The concerns regarding security vulnerabilities, legal uncertainty, technical instability, and potential agent-layer lock-in are legitimate, but ultimately represent engineering and governance challenges rather than insurmountable obstacles.

The primary advantages, namely reduced integration costs, enhanced competition through lower switching barriers, and democratised automation, offer substantial economic and social benefits that traditional interoperability efforts have pur-

sued but never fully achieved. Startups and small teams can connect to major services almost instantly; enterprises gain resilience through automatic failover to alternate providers; and individual users regain control over their data and workflows. More importantly, this development might already be present in its early form, and several issues with unauthorised AI-mediated agent interactions have already surfaced. WIRED and Forbes have accused Perplexity AI of violating, respectively, robots.txt (Mehrotra & Marchman, 2024) and copyright laws (Lane, 2024; Shrivastava, 2025). Some open-source AI crawlers (e.g., Firecrawl and Botright) include CAPTCHA bypass mechanisms that may violate terms of service. The OpenAI-powered spambot Akirabot has successfully spammed over 80,000 websites, using LLM-written messages and CAPTCHA-bypassing mechanisms (Delamotte & Walter, 2025).

As agents become more effective and scalable, this trend will likely intensify: any agent workflow interacting with APIs or websites indirectly contributes to universal interoperability. The Internet risks becoming an intricate, interconnected network of AI systems that interact with minimal human oversight. Therefore, the strategic question is not whether AI-mediated interoperability will emerge, but whether it develops in a principled, secure fashion or through uncoordinated implementations with varying reliability and security.

We advocate a proactive approach: establishing lightweight frameworks and conventions now, while the agent ecosystem is still relatively immature. By implementing appropriate guardrails for security, permission management, and interface standardisation, we can direct this capability toward beneficial outcomes while mitigating potential harms. Universal interoperability offers a technological path to more competitive digital markets that complements regulatory interventions, potentially enabling new use cases across domains where data silos have hindered progress.

In summary, while universal interoperability introduces new challenges, these represent addressable problems that the ML community is well-positioned to solve. The following section outlines specific directions for addressing the most critical challenges.

## 5. Call to Action

Universal interoperability requires solving unique technical and governance challenges that exceed the scope of this paper or any single researcher's effort. In this section, we highlight open research questions, early prototypes, and areas where the ML community can make high-impact contributions. We discuss three main areas, namely **agent-friendly interfaces**, **security by design**, and **ecosystem infrastructure**.

## 5.1. Agent-Friendly Interfaces

Current APIs and websites are designed for either human users or traditional clients, forcing LLM agents to infer missing context. An agent presented with an OpenAPI schema must interpret implicit business rules that would be assumed to be known to developers. Without additional guidance, agents rely on trial and error (submitting requests, handling failures, and adjusting prompts) (Song et al., 2024), which reduces reliability and increases integration time.

Adding minimal metadata beyond field descriptions can provide the rationale and implicit knowledge for each endpoint. Since this information is in plain text, non-technical users or even LLM can write it. At its simplest, the metadata could be a link to human documentation or any helpful resources that provide the implied context (blog posts, official website, forums...). At its most advanced, service providers could offer an LLM-based endpoint where agents can request schema clarifications. Regardless of the specific approach, this information would help agents avoid guesswork and handle edge cases more predictably. We report (Bandlamudi et al., 2025) as an early contribution in this direction.

Web pages present similar challenges: an agent must parse the DOM or rendered HTML to locate elements and execute workflows. Embedding a small manifest can annotate form fields, buttons, and links with corresponding API calls or structured action identifiers. For example, a checkout page could label its "Submit Order" button with the endpoint `POST /api/order` and list required parameters. Agents can then bypass the UI by calling the API directly and following documented schemas, simplifying error handling and avoiding brittle UI parsing. `llms.txt` (Howard, 2024), which provides an LLM-friendly explanation of a web page, represents an early step in this direction.

These metadata extensions would build on existing standards, requiring only the addition of links or annotations, rather than adopting a new specification. The research question concerns how much detail is necessary for robust agent operation, and what combination of static metadata and dynamic explanation services provides the best balance of effort and reliability.

## 5.2. Security by Design

Current security mechanisms in LLM agent frameworks inadequately protect not only users (Kim et al., 2025), but also the websites these agents access. While API endpoints typically implement standardised permission and rate limit systems (Hardt, 2012; Serbout et al., 2023), web pages lack standardised ways to specify which UI actions an agent can perform and how often. This gap may compel website owners to introduce blanket bans on AI agents, limiting interoperability. Therefore, it is essential to develop permission systems that specify agent permissions, frequency limits, and delegation authority. This requires adapting multiple security components for AI agents, including ID systems (Chan et al., 2024), authenticated delegation mechanisms (South et al., 2025), data usage rights, and rate limits.

For users operating in untrusted settings, validating agent behaviour before using production data requires more than static analysis. Research prototypes such as ToolEmu (Ruan et al., 2024), which simulates external APIs, and AgentSims (Lin et al., 2023), which creates synthetic task environments, represent promising steps toward secure testing frameworks. Systems like SandboxEval (Rabin et al., 2025) run agents in isolated containers to detect privilege-escalation attempts. However, much remains to be done to prevent adversarial attacks against agents (Wu et al., 2024). Adding agent-focused test suites to continuous integration pipelines and developing human-in-the-loop review systems represent key areas for future research.

That said, even the best pre-deployment checks cannot guarantee ongoing security. Continuous monitoring solutions, such as runtime policy enforcement layers (Kim et al., 2025), are emerging. Concretely, we envision runtime enforcement as a three-layer architecture. First, signed permission documents (South et al., 2025) specify allowed endpoints, data-use policies, rate limits, and delegation authority for each agent. Second, a runtime policy checker verifies each action against the permission document before execution, blocking or escalating violations. Third, an automatic rollback (or kill-switch) layer can reverse (or halt) agent activity when the monitoring system detects out-of-policy behaviour. The main challenge here is the policy-checker layer: implementing per-action policy verification at low enough latency that it does not bottleneck agent workflows, and with low enough false-positive rates that it does not require constant user intervention. The ML community can contribute through learned policy classifiers, efficient symbolic checkers, or hybrid approaches.

## 5.3. Ecosystem Infrastructure

Interoperability requires open protocols that enable agent systems to discover each other, delegate tasks, and share data securely. Two prominent (and complementary) proposals have emerged from leading AI companies: Google's Agent-to-Agent (A2A) protocol (Google, 2025), which defines a framework with discovery flows, authentication, and structured message schemas, and Anthropic's Model Context Protocol (MCP) (Anthropic, 2024), which standardizes

how agents connect to external tools, data sources, and workflows. While both are open source, their origin within single companies raises concerns about future fragmentation or lock-in[1]. To prevent such outcomes, we advocate active participation in multi-stakeholder working groups and standards bodies such as the W3C AI Agent Protocol community group, the Lightweight Agent Standards Working Group, the NANDA ecosystem and Eclipse LMOS, where academic, commercial, and open-source stakeholders can jointly maintain specifications. As a practical safeguard, agent frameworks should also support adapters that translate between A2A, MCP, and any emergent protocols, ensuring that agents remain interoperable even in the case of ecosystem fragmentation and can switch to alternative protocols if needed.

A second pillar of robust infrastructure is managing technical debt arising from AI-generated integration code. Unreviewed prompts and automatically synthesized scripts can create hidden dependencies, obscure bugs, and brittle workflows. To address this, the community should curate and maintain reference implementations of common integration patterns, covering authentication, pagination, error handling, and rate-limit compliance, and make these templates publicly available under permissive licenses. API providers can assist by publishing machine-readable changelogs (for example, an OpenAPI diff format), which LLM-based tools can scan to detect breaking changes in downstream adapters. Each generated connector should also include version information and metadata about its creation process. In the long term, agents themselves could propagate updates and automatically adapt or deprecate outdated adapters, thus keeping the ecosystem in sync without central coordination.

Finally, the ecosystem must prevent hidden biases and anti-competitive behaviours at the model and framework levels (e.g., favouritism towards specific services with commercial ties to the agent developers) (Kapoor et al., 2025). Transparent monitoring is crucial: agents should log their decision process, including which services were evaluated and the selection criteria, and present accessible audit trails. Above all, open-source agent frameworks and models represent the best defense against vendor capture. By allowing inspection of training data, prompt logic, and integration policies, we can ensure that agents serve user interests rather than hidden commercial objectives.

---

[1]Though it should be noted that some progress has been made towards decentralisation, such as Google announcing that the Linux Foundation will steward A2A, and Anthropic allowing external contributions to MCP.

## 6. Alternative Views

To provide a complete analysis, we review four prominent alternative perspectives and examine how each can inform or be integrated into our proposed solutions.

**Regulation-first perspective** It can be argued that technical workarounds, such as AI-mediated scraping or UI automation, merely delay the real solution: mandatory open API requirements. Without binding obligations, platforms have strong incentives to maintain proprietary barriers whenever it is economically advantageous (Bourreau et al., 2022). Until such issues are addressed, user-driven agents will operate in a legal gray area and be subject to cease-and-desist orders of questionable authority. However, regulatory processes tend to be slow and reactive (Afina et al., 2024): by the time new rules are in place, dominant ecosystems may already have cemented their positions. AI-mediated interoperability can provide portability tools today, implementing data portability principles while regulators catch up. That said, we fully agree that technical solutions must be complemented by robust legal frameworks: agent interfaces should incorporate regulatory requirements (e.g., explicit consent models, data-use restrictions), and standards bodies should collaborate with lawmakers to ensure both technical soundness and legal compliance.

**Ontologies over bespoke formats** Another perspective emphasizes that reliable data exchange requires formally verified ontologies rather than improvised schema mappings. Semantic Web research shows how shared registries (e.g. `schema.org`) and rigorous schema validation can prevent semantic drift (Şimşek et al., 2018). However, the slow pace of standardisation and the expertise required for ontology engineering have limited widespread adoption of these technologies (Neuhaus & Hastings, 2022). A middle ground is to employ established ontologies where feasible while using LLMs to bridge gaps or handle edge cases.

**Security-first caution** From a security standpoint, powerful AI agents resemble unsupervised code execution with the potential to leak data, exploit vulnerabilities, or automate attacks at scale. One option is to limit agents to narrow, human-approved tasks with explicit checkpoints before external interactions. However, such severe constraints would undermine the flexibility and efficiency benefits of AI-mediated integration. A more balanced approach is to use a mix of defenses: agents a) operate under signed permission documents that specify allowed endpoints, rate limits, and data-use policies; b) run in isolated sandboxes; and c) are overseen by continuous monitoring systems. This model preserves strong oversight while enabling automated integration.

**Economic sustainability** Another concern is that platforms must control API access to recover infrastructure costs and maintain service quality. Unrestricted AI traffic can create abusive usage patterns (Kong Inc., 2024), reduce advertising revenue (Ryan, 2024), and bypass paid APIs. We agree with these concerns: interoperability tools should respect the same billing, rate-limiting, and contractual frameworks that human developers follow, and the decline of user traffic might require new business models. By making rate limits and usage fees explicit in permission documents and supporting tiered APIs, platforms can balance user empowerment with sustainability, ensuring that openness does not compromise service quality or business viability.

## 7. Conclusion

Universal interoperability via LLM-based agents provides a mechanism to reduce integration effort, lower barriers to multi-homing, and democratise automation. By converting complex integration projects into prompt-based interactions, this approach can generate significant cost savings, enhance competition, and expand access to new applications. At the same time, we have identified several challenges: security vulnerabilities, legal uncertainty, unpredictable agent behaviour, accumulating technical debt, and the risk of agent-layer lock-in. While these issues require mitigation, they are far from insurmountable and can be addressed through thoughtful engineering and governance.

To that end, we have outlined three foundational pillars: agent-friendly interfaces, security by design, and ecosystem infrastructure. These technical measures require complementary engagement with regulators and standards bodies to establish explicit consent models and data-use constraints within agent interfaces.

We invite the ML research community to contribute to this infrastructure, for example by developing and publishing permission manifests and policy checkers, creating and maintaining open-source adapter libraries, documenting interface metadata, and designing rigorous evaluation suites and certification processes. In the near term, these efforts will produce more secure and reliable agent workflows. Over time, they can help dismantle entrenched silos, empower users with genuine data portability, and support resilient, competitive digital markets. By establishing this foundation now, while AI agent technologies are still being developed, we can steer interoperability toward secure, equitable, and sustainable outcomes.

## Acknowledgments

This work was supported by the EPSRC Centre for Doctoral Training in Autonomous Intelligent Machines and Systems n. EP/Y035070/1, in addition to Microsoft Ltd. We would like to thank Emanuele La Malfa and Leslie Tao.

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
