# OpenReview forum: "Position: LLM Agents Are the Antidote to Walled Gardens"
_ICML.cc/2026/Position_Paper_Track — ICML 2026 Position Paper Track regular_

### Official Review · Reviewer_LEZ6 · 2026-02-16

**Significance:** 3
**Argument Clarity:** 3
**Rating:** 4
**Confidence:** 3

**Questions:**

1. Regarding agent friendly interfaces, what is the minimum set of metadata required to significantly reduce agent hallucination compared to raw html or openapi schemas?

2. How do you propose handling the incentive mismatch where platforms might deliberately design interfaces to be agent hostile to protect revenue even if metadata standards exist?

3. Can you elaborate on the mechanisms for runtime policy enforcement that would prevent an agent from inadvertently violating legal or safety constraints while attempting an integration?

**Alternative Views Section:**

Yes

**Compliance With Llm Reviewing Policy A Conservative:**

Affirmed.

**Discussion Potential:**

2

**Paper Summary:**

This position paper argues that large language model agents serve as a fundamental remedy for the closed application layer of the modern internet. The authors observe that while core internet protocols are open, user facing services have become walled gardens due to the high cost of maintaining integrations and strategic platform lock in. The core thesis is that agents capable of automated translation between api formats and robust interaction with human oriented user interfaces make interoperability dramatically cheaper and effectively unavoidable. This shift is defined as universal interoperability. The paper acknowledges significant risks including new forms of technical debt, legal uncertainties, and security vulnerabilities like adversarial prompt injection. To address these, the authors propose a call to action centered on three pillars: developing agent friendly interfaces with minimal metadata, implementing security by design through isolated sandboxes and permission manifests, and building ecosystem infrastructure via open protocols and reference implementations.

**Position:**

Yes

**Position In Title:**

Yes

**Related Work:**

3

**Strengths And Weaknesses:**

The paper addresses a highly relevant and timely topic for the machine learning community by connecting the technical capabilities of agents to broader economic and structural shifts in digital markets. A major strength is the clear articulation of why agents are qualitatively different from previous middleware or robotic process automation, specifically highlighting their ability to infer semantic correspondences and adapt to changing interfaces at runtime. The distinction between syntactic and semantic interoperability is well grounded in established literature. Furthermore, the paper provides a balanced view by dedicating significant space to risks such as silent data corruption from hallucinations and the potential for new monopolies at the agent framework layer. The call to action is concrete, offering specific research directions like developing machine readable changelogs for apis and agent focused test suites.

However, the paper is somewhat speculative regarding the legal landscape, noting that agents currently operate in a gray area without providing a clear path for how technical standards might influence or resolve terms of service conflicts. While the alternative views section is present and addresses economic sustainability, it could more deeply explore the potential for a catastrophic breakdown of service quality if platforms face massive unmonetized agent traffic. Additionally, the reliance on emergent protocols like a2a and mcp as foundational infrastructure is noted, but the paper could more critically evaluate how to ensure these remain truly multi stakeholder rather than just open source in name.

**Support:**

3

---

> ### Author Rebuttal · Authors · 2026-03-31
>
> Dear Reviewer LEZ6,
>
> Thank you for your detailed engagement with the paper and for the specific, targeted questions. We address the identified weaknesses and each question below.
>
> **On the legal landscape.** We agree that the paper can benefit from analyzing how technical standards might interact with terms-of-service conflicts. We propose adding a discussion that clarifies our view: technical standards such as permission manifests and metadata annotations do not resolve legal disputes directly, but they make agent behaviour auditable and machine-readable, which makes legal adjudication more tractable. The analogy is robots.txt, which began as an informal convention and evolved into a legally referenced standard that courts and regulators now treat as a baseline for acceptable crawler behaviour. Permission documents for agents could follow a similar trajectory. We can also connect this more explicitly to existing legal instruments: GDPR Article 20's portability provisions and the DMA's interoperability obligations both assume machine-readable interfaces, and agent-facing metadata standards would operationalize these requirements.
>
> **On unmonetized agent traffic.** We propose expanding the discussion of economic sustainability in Section 6 to address three points:
>
> 1. The precedent of search engine crawling, where an initial period of unregulated access (which created concerns about server load and bandwidth costs from automated traffic) eventually gave way to cooperative frameworks (e.g. robots.txt).
> 2. The observation that agent traffic is not inherently unmonetizable (agents that purchase services, complete transactions, or interact with paid API tiers represent demand that platforms lose by blanket blocking).
> 3. The design of tiered access models that distinguish between read-only crawling and transactional agent interactions, with different rate limits and billing structures for each.
>
> **On A2A/MCP governance.** We share your concern. The paper's advocacy for adapter layers that translate between A2A, MCP, and emergent protocols is precisely to defend against single-company capture: if agents can switch protocols, no single specification achieves lock-in. We propose adding a more explicit discussion of governance mechanisms (e.g., charter requirements and voting structures) to define more clearly what we mean by “multi-stakeholder”.
>
> **Q1: Minimum metadata for reducing hallucination.** This is an open empirical question that the community should prioritize. The natural experimental design is an ablation study comparing agent task success rates across increasing levels of metadata richness: raw HTML alone, OpenAPI schema without additional context, OpenAPI augmented with natural-language information for each endpoint (e.g., business rules, expected workflows, common edge cases), and full metadata including example request/response pairs and links to human documentation. Bandlamudi et al. (2025), already cited in the paper, represent an early step in this direction, showing that even minimal additional context about API semantics substantially improves agent reliability.
>
> **Q2: Incentive mismatch and agent-hostile design.** There are three considerations that suggest that deliberate agent hostility is a losing long-term strategy. First, the most effective anti-agent measures (aggressive CAPTCHAs, interface obfuscation, decoy pages) also degrade the human user experience; Searles et al. (2023), cited in the paper, document how reCAPTCHA already frustrates legitimate users. Second, the arms race favors agents: LLM capabilities improve faster than obfuscation techniques, as evidenced by Plesner et al. (2024, see paper) showing high success rates against reCAPTCHAv2. Third, and perhaps most importantly, rejecting agent traffic leaves money on the table: in verticals where the platform sells services (e.g., e-commerce, SaaS, travel booking), agents completing purchases or subscribing to paid API tiers represent monetizable demand.
>
> **Q3: Runtime policy enforcement.** The paper discusses several components in Section 5.2; we can consolidate these into a clearer layered architecture. The first layer consists of signed permission documents specifying allowed endpoints, rate limits, data-use policies, and delegation authority. The second layer is a runtime monitoring system that checks each agent action against the permission document before execution, flagging or blocking violations. The third layer provides automatic rollback or escalation when an agent operates outside its defined constraints. The key challenge, which we flag in Section 5.2, is implementing these checks without introducing prohibitive latency or false positives. This is an area where the ML community can contribute through efficient policy-checking mechanisms.
>
> We hope these responses and proposed revisions address your concerns. Let us know if you have any other questions.
>
> Best,
>
> The Authors

---

### Official Review · Reviewer_nGTD · 2026-03-03

**Significance:** 3
**Argument Clarity:** 3
**Rating:** 3
**Confidence:** 4

**Questions:**

Q1:  While the concept of agents as universal adapters is compelling, the paper currently lacks sufficient proof of its viability in large-scale, real-world environments. I recommend adding empirical data, case studies, or a detailed qualitative analysis that addresses how these agents handle scale-specific challenges.

Q2:  The position paper would benefit from a more structured formulation of the trade-offs associated with universal interoperability. Could the authors formalize the balance between its advantages and potential risks? Furthermore, identifying specific domains, such as healthcare, finance, or open-source development, where the benefits definitively outweigh the risks would provide highly valuable guidance for practical deployment.

**Alternative Views Section:**

Yes

**Compliance With Llm Reviewing Policy A Conservative:**

Affirmed.

**Discussion Potential:**

2

**Paper Summary:**

The authors argue that LLM agents introduce new risks because they can automatically translate across disparate data schemas and interact directly with GUIs, eliminating the need for official APIs. They further contend that AI-mediated adapters make system integration inexpensive and effectively unavoidable. The paper closes with a call to action for the ML community to build agent-friendly interfaces, robust security guardrails, and open ecosystem infrastructure.

**Position:**

Yes

**Position In Title:**

Yes

**Related Work:**

3

**Strengths And Weaknesses:**

Pros:
* The framing of LLM agents as _universal adapters_ is intuitive and provides an interesting lens for understanding the future of web automation and API integration.
* The call to action section provides concrete, pragmatic directions for the ML community, such as developing metadata standards and robust permission manifests.

Cons:
* Because this is a position paper, the core claims are not strongly supported by evidence. The authors assume that agents can reliably and affordably act as universal adapters at scale, but they do not provide quantitative data or supporting evidence.
*  While the authors attempt to examine the key issue of universal interoperability, the core position of the paper may already be relatively well-known and accepted within the broader machine learning and developer communities.

**Support:**

2

---

> ### Author Rebuttal · Authors · 2026-03-31
>
> Dear Reviewer nGTD,
>
> Thank you for your constructive engagement with the paper and for the specific, actionable suggestions in Q1 and Q2. We address your concerns below.
>
> **On novelty.** You suggest that the core position may already be well-known within the ML community. We partially agree: the observation that LLM agents can interact with APIs and GUIs is widely recognised. However, our contribution does not lie in this observation but in its consequences. The paper formally defines universal interoperability, connects it to the economics of network effects and switching costs, analyzes its interaction with ongoing regulatory efforts (the DMA, GDPR, and the ACCESS Act), and identifies the infrastructure that the ML community must build to ensure this shift develops in a principled fashion. This type of synthesis has not, to our knowledge, been presented elsewhere.
>
> **On empirical evidence.** The paper includes significant evidence: Section 4.3 documents real-world instances of AI-mediated interoperability (AkiraBot spamming over 80,000 websites via LLM-generated messages, Perplexity AI violating robots.txt, and open-source crawlers shipping with mechanisms to bypass CAPTCHAs), while Sections 3.1-3.2 cite specific systems and benchmarks (RestGPT, Gorilla, ReAct on WebArena and MiniWoB). To strengthen the quantitative grounding, we can add trajectory data: on WebArena, a realistic benchmark of multi-step web tasks, LLM agents improved from approximately 14% task success (2023) to 55-60% (2025), which is roughly a 4-time improvement in two years. Additionally, more and more frontier companies have introduced their own web agents and agent browsers, such as ChatGPT Agent \[1\], OpenAI Atlas \[2\], Claude in Chrome \[3\], Manus \[4\], and Perplexity Comet \[5\]. Traditional API integration, by contrast, requires hundreds of engineering hours per integration due to needing to map schemas, develop client libraries, test compatibility, and perform ongoing maintenance \[6\]. The gap between these two regimes supports the "cheap and inevitable" claim with existing evidence.
>
> **On scale-specific challenges (Q1).** We agree that a more detailed analysis of how agent-mediated integration behaves at scale would strengthen the paper. The camera-ready version will include a discussion of:
>
> * How failures cascade across chained adapters, where errors in one agent-mediated link propagate through downstream integrations;
> * Latency and efficiency overhead of UI-based versus API-based agent interactions;
> * Compounding hallucination risk in multi-hop schema translations, where each intermediate mapping introduces opportunities for silent data corruption.
>
> Quantifying these rates and developing mitigation strategies (e.g., intermediate validation checkpoints or schema-aware self-correction) represents an open research direction we propose to highlight in the revised paper.
>
> **On domain-specific trade-offs (Q2).** We find this suggestion particularly valuable. The paper already discusses domain-specific interoperability standards (e.g., HL7 FHIR in healthcare and ISO 20022 in financial messaging), but does not systematically analyze how agent-mediated interoperability interacts with them. We propose adding a structured comparison across domains, covering the current status of interoperability, the benefits that agent-mediated bridging provides, as well as the main risks. In healthcare, for example, agents could act as bridges between FHIR-compliant and legacy non-FHIR systems, which would improve data portability but would increase risks around patient data privacy and regulatory compliance; in finance, agents could mediate between ISO 20022 and legacy SWIFT message formats, which would reduce friction but make audit trails more complex.
>
> We hope that these clarifications and proposed revisions address your concerns. Let us know if there are further analyses or specific evidence that would increase your confidence in the paper.
>
> Thanks again\!
>
> The Authors
>
> \[1\] OpenAI. “ChatGPT Agent.” *OpenAI Help Center*, help.openai.com/en/articles/11752874-chatgpt-agent.
>
> \[2\] OpenAI. “Introducing ChatGPT Atlas.” *OpenAI*, 21 Oct. 2025, openai.com/index/introducing-chatgpt-atlas/.
>
> \[3\] Anthropic. “Get Started with Claude in Chrome.” *Claude Help Center*, support.claude.com/en/articles/12012173-get-started-with-claude-in-chrome.
>
> \[4\] Manus. “Introducing Manus Browser Operator.” *Manus*, 18 Nov. 2025, manus.im/blog/manus-browser-operator.
>
> \[5\]  “Comet Browser: a Personal AI Assistant.” *Perplexity*, perplexity.ai/comet/.
>
> \[6\] Knoche, Holger, and Wilhelm Hasselbring. "Continuous api evolution in heterogenous enterprise software systems." 2021 IEEE 18th International Conference on Software Architecture (ICSA). IEEE, 2021\.

---

> > ### Author Rebuttal · Reviewer_nGTD · 2026-04-03
> >
> > Thank you to the authors for the thoughtful rebuttal. I have read the response carefully. The rebuttal has improved my understanding of the paper and addressed several of my concerns. I still remain concerned that the paper’s central claims are not yet supported by sufficiently strong empirical backing or rigorous theoretical modeling. The additional examples, benchmark references, and anecdotal cases cited in the rebuttal are helpful for illustrating plausibility, but they do not yet amount to a systematic empirical foundation for the stronger claims about large-scale viability, affordability, or inevitability.

---

### Official Review · Reviewer_oUBd · 2026-03-12

**Significance:** 2
**Argument Clarity:** 2
**Rating:** 2
**Confidence:** 3

**Questions:**

See the strengths and weaknesses part.

**Alternative Views Section:**

Yes

**Compliance With Llm Reviewing Policy A Conservative:**

Affirmed.

**Discussion Potential:**

2

**Paper Summary:**

This paper argues that LLM agents can dismantle the walled gardens of closed digital platforms by making data integration cheap and inevitable. The authors introduce universal interoperability, where AI agents autonomously translate between different APIs and navigate user interfaces, bypassing traditional technical barriers. While this shift promises to increase market competition, drastically reduce engineering costs, and enhance data portability, it also introduces significant risks like security vulnerabilities and technical debt. Consequently, the authors urge the machine learning community to proactively develop agent-friendly interfaces, secure-by-design frameworks, and open ecosystem infrastructures.

**Position:**

Yes

**Position In Title:**

Yes

**Related Work:**

2

**Strengths And Weaknesses:**

Strength
===

1. The paper effectively highlights the unique advantages of LLM agents and the massive changes they introduce to the digital landscape. By pointing out that agents can automatically translate between data formats and interact with interfaces designed for humans, the authors convincingly argue that interoperability is becoming both drastically cheaper and effectively unavoidable.
2. A standout feature of this work is its holistic analysis. The authors successfully bridge computer science with industrial organization economics—discussing concepts like network effects, switching costs, and multi-homing —while also contextualizing the technical problem within current legal and regulatory interoperability efforts.
3. The paper does a good job of proposing several areas worthy of future investigation. Categorizing the required efforts into agent-friendly interfaces, security by design, and ecosystem infrastructure provides a structured roadmap for the community.

Weakness
===

The core weakness of this position paper is its reliance on high-level technological forecasting at the expense of grounded scientific analysis. The authors do not adequately critique the methodological, algorithmic, or theoretical shortcomings of existing LLM agent research. The overarching claims regarding the imminent feasibility and scalability of universal interoperability lack empirical backing, rigorous theoretical modeling, or quantitative projections to substantiate the position.

While the identified research directions are practically relevant, the Call to Action reads more like a policy manifesto or industry strategy than a rigorous academic research agenda. For example, proposing the addition of plain-text metadata to APIs or advocating for multi-stakeholder working groups  are organizational and engineering solutions. The paper fails to deeply analyze the underlying fundamental machine learning challenges that are scientifically necessary to build this proposed ecosystem.

**Support:**

2

---

> ### Author Rebuttal · Authors · 2026-03-31
>
> Dear Reviewer oUBd,
>
> Thank you for your review and for acknowledging the paper's strengths (in particular, the holistic analysis, the structured roadmap, and the useful framing of how LLM agents reshape the digital landscape). We address the two main concerns below.
>
> **On "high-level technological forecasting" vs. grounded analysis**
>
> As a position paper, our contribution is the argument and research agenda rather than novel algorithms or empirical results. That said, we want to push back on the characterization of our paper as “technological forecasting”. The phenomena we describe are documented and ongoing: Section 4.3 cites several concrete instances, such as AkiraBot spamming over 80,000 websites using LLM-generated messages, Perplexity AI being accused of violating robots.txt, and open-source crawlers shipping with CAPTCHA bypass mechanisms. These are current, real-world examples of AI-mediated interoperability occurring without coordination or safeguards, which is precisely the gap our paper identifies.
>
> The trajectory of agent capabilities further supports our argument. On WebArena, which we cited as a realistic benchmark of multi-step web tasks, LLM agents improved from \~14% success (2023) to \~55-60% (2025), which is roughly a 4x improvement in two years. Another piece of evidence is the proliferation of frontier AI web agents, such as ChatGPT Agent \[1\], OpenAI Atlas \[2\], Claude in Chrome \[3\], Manus \[4\], and Perplexity Comet \[5\]. Meanwhile, integrating traditional APIs often requires hundreds of engineering hours, including mapping schemas, testing compatibility, working on client libraries, and ongoing maintenance (as discussed in Section 2 and supported by \[6\]). The contrast between this engineering effort and using an agent (which only requires a prompt) is the core of our "cheap and inevitable" claim, and it is grounded in existing evidence.
>
> **On the Call to Action as "policy manifesto"**
>
> We understand the concern that some of our proposed directions (e.g., metadata annotations, multi-stakeholder working groups) appear more organizational than scientific. However, we note two things. First, many foundational contributions in systems and ML began as position-level framings of what needs to be built: the Semantic Web stack, OAuth, and OpenAPI all started as conceptual proposals before becoming engineering standards. Position papers serve precisely this function, which is to identify the research agenda.
>
> Second, the paper does identify concrete ML research challenges, though we acknowledge they could be emphasized more explicitly. We would be happy to add a dedicated subsection or table summarizing the open ML problems discussed in Sections 4.2 and 5, including:
>
> 1. Adversarial robustness of web agents to malicious pages and prompt injection;
> 2. Hallucinations of schema mappings and silent data corruption;
> 3. Prompt sensitivity;
> 4. Formal verification of LLM-generated glue code;
> 5. Detection and mitigation of schema drift in interfaces;
> 6. Preventing agent-layer lock-in through open, auditable model behavior.
>
> Each of these represents an important problem that requires both advances in ML and engineering effort.
>
> We believe that these clarifications, together with the proposed revision, show that the paper meets the standard for an ICML position paper: a clearly argued, well-grounded thesis that identifies an important research agenda for the community.
>
> Thank you again, and let us know if you have any other questions\!
>
> The Authors
>
> \[1\] OpenAI. “ChatGPT Agent.” *OpenAI Help Center*, help.openai.com/en/articles/11752874-chatgpt-agent.
>
> \[2\] OpenAI. “Introducing ChatGPT Atlas.” *OpenAI*, 21 Oct. 2025, openai.com/index/introducing-chatgpt-atlas/.
>
> \[3\] Anthropic. “Get Started with Claude in Chrome.” *Claude Help Center*, support.claude.com/en/articles/12012173-get-started-with-claude-in-chrome.
>
> \[4\] Manus. “Introducing Manus Browser Operator.” *Manus*, 18 Nov. 2025, manus.im/blog/manus-browser-operator.
>
> \[5\]  “Comet Browser: a Personal AI Assistant.” *Perplexity*, perplexity.ai/comet/.
>
> \[6\] Knoche, Holger, and Wilhelm Hasselbring. "Continuous api evolution in heterogenous enterprise software systems." 2021 IEEE 18th International Conference on Software Architecture (ICSA). IEEE, 2021\.

---

### Official Review · Reviewer_X6cw · 2026-03-13

**Significance:** 3
**Argument Clarity:** 4
**Rating:** 5
**Confidence:** 4

**Questions:**

Given the fair proliferation of agentic LLM systems, the authors can further augment the paper with a concrete case study of specific domains and demonstrate how the local systems have become interoperable in a lasting (and not just transient) way?
And similarly a strong rebuttal to this paper would be for a reviewer to demonstrate that even though this is true - certain industries have NOT become interoperable. For example, in finance, what can be done, and what cannot be done is sufficiently described in policy that the there is very little "interoperability" even when the technical reasons to do so are somewhat easy. I would be curios to hear the authors rebuttal to this!

**Alternative Views Section:**

Yes

**Compliance With Llm Reviewing Policy A Conservative:**

Affirmed.

**Discussion Potential:**

4

**Paper Summary:**

This paper presents universal interoperability. The authors argue that traditional system have led to incentives where the market leaders have little to no-incentive in making their APIs and software interoperable. However, LLM agents with their ability to easily convert between data formats AND operate UI designed for humans, makes the future of interoperable interfaces "inevitable".

**Position:**

Yes

**Position In Title:**

Yes

**Related Work:**

3

**Strengths And Weaknesses:**

Strength
+ The paper presents an interesting topic for the following reason - with the advents of LLMs, it was generally believed that the internet would become a walled garden - due to incentives to prevent LLM providers from accessing data. However, this paper argues that the fact that LLM agents are so omni-capable that, we are instead seeing the opposite - wider open gardens. This contradiction, is simple, and makes for a great argument.
+ This paper presents a thorough analysis  that is not just grounded in LLM developments, but also tackles classical literature in industrial organization (Katz and Shapiro), and includes regulatory perspective.

Weakness
- The biggest weakness of the paper is it's lack of comment on policy. In-fact the first occurrence of policy (NOT about how it's  enforced, but rather what happens when it is violated), is in Section 5.2. While I agree with all the comments on market incentives, and the capabilities of LLM Agents, a core fundamental for trade is - agreement with terms of service. This is the same reason, an enterprise can choose to do something or not - since the dis-incentive of not following policy is enforceable through law. So, in such a system if the leading providers have an incentive to keep their systems NON-interoperable, can't they simple express it as a policy, and all LLM agents are required to adhere to it - or risk violating the policy and face consequences? In such a scenario, would "universal interoperability" be "inevitable"?
- The paper also makes an assumption that when LLM agents make a set of systems interoperable, these are sufficiently shared/reproducible? Would it be possible that certain interoperable systems are in and of itself sufficiently complex - that the interoperability would become a "smaller" although much powerful walled garden?

**Support:**

4

---

> ### Author Rebuttal · Authors · 2026-03-31
>
> Dear Reviewer X6cw,
>
> Thank you for your review and for your positive assessment\! We address both weaknesses and the question below.
>
> **On policy enforcement as a hard barrier (W1).** You raise the strongest version of the regulation-first perspective from Section 6: if platforms can prohibit agent-mediated access in their terms of service and enforce this through law, the "inevitable" claim may not hold. We respond on three levels.
>
> First, the gap between policy and enforcement is wide. While terms of service routinely prohibit automated scraping (see Fiesler et al., 2020 from the paper), enforcement remains inconsistent and the legal landscape is unsettled (see Atkinson, 2025). The evidence in Section 4.3 is instructive: Perplexity AI has been accused of violating robots.txt, AkiraBot spammed over 80,000 websites using LLM-generated messages and CAPTCHA bypass, and open-source crawlers ship CAPTCHA-bypass mechanisms as default features. In practice, ToS prohibitions have not contained agent-mediated access.
>
> Second, even in heavily regulated domains, the regulatory trajectory favors *mandated* interoperability. For example, the DMA requires gatekeepers to interoperate on core messaging functionalities, Open Banking (UK) and PSD2 (EU) mandate API access in finance, and the 21st Century Cures Act prohibits information blocking in US healthcare. In short, policy is increasingly accelerating interoperability rather than preventing it. The relevant question is not whether platforms can prohibit agent access, but whether the broader regulatory environment will continue to support such prohibitions when interoperability is itself becoming a policy objective.
>
> Third, our "inevitability" claim does not require that agents forcibly open every domain or platform. It requires that the cost structure of interoperability has shifted such that the equilibrium moves: platforms that remain closed now bear higher relative costs than before, including lost agent-mediated revenue, increased scrutiny, user attrition, and the UX degradations required to block agents.
>
> We agree that the paper underweights the consequences of policy violation. We propose discussing enforcement mechanisms and legal risk in Section 4.2 as a distinct risk category rather than deferring the discussion to Section 5.2.
>
> **On agent-layer lock-in (W2).** We agree that complex interoperability solutions could themselves become a smaller but more powerful walled garden, and we discuss the risk in Section 4.2. Our primary countermeasure is architectural: the protocol-agnostic adapter layers advocated in Section 5.3, which translate between A2A, MCP, and emergent protocols, ensure that no single specification achieves lock-in. Combined with open-source agent frameworks, this makes the interoperability layer itself interoperable. However, this outcome is not guaranteed and will require sustained community effort, which is precisely why the call to action matters.
>
> **On the finance case study (Q).** Finance is the strongest counterexample to our inevitability claim, as it’s heavily regulated, with real legal consequences for violations and incumbents with strong incentives to resist openness. Yet the trajectory is instructive. To continue our above discussion on interoperability, Open Banking (UK) and PSD2 (EU) required banks to expose payment account APIs, creating entire fintech ecosystems that did not exist before. The remaining friction is mostly due to infrastructure rather than policy \[1, 2\]: the adoption of ISO 20022 is incomplete, and SWIFT message formats still dominate many cross-border channels. Agents (deployed with the right security and auditability infrastructure, of course) could bridge between ISO 20022 and legacy formats, extending interoperability beyond the currently mandated scope, and the reduction in engineering costs would make this feasible for smaller fintech companies that cannot currently afford the integration overhead. We propose adding this as a concrete case study demonstrating how agent-mediated interoperability interacts with, rather than circumvents, existing regulatory frameworks.
>
> Thank you again for your review, and let us know if you have any further questions\!
>
> The Authors
>
> \[1\] Weißkopf, Ingrid, and Roland Nehl. "ISO 20022 Migration: Where We Stand and the Road Ahead." *The Paypers*, 24 Sept. 2025, thepaypers.com/regulations/interviews/iso-20022-migration-where-we-stand-and-the-road-ahead.
>
> \[2\] Dann, Clarissa. "Embracing ISO 20022 for a Connected Future." *flow*, Deutsche Bank, 7 Nov. 2024, flow.db.com/topics/cash-management/embracing-iso-20022-for-a-connected-future.

---

> > ### Author Rebuttal · Reviewer_X6cw · 2026-04-04
> >
> > Thank you for the rebuttal.
> >
> > > We agree that the paper underweights the consequences of policy violation. We propose discussing enforcement mechanisms and legal risk in Section 4.2 as a distinct risk category rather than deferring the discussion to Section 5.2.
> >
> > I would recommend NOT just in section 4.2, but I think this is an interesting argument that merits to be surfaced in the introduction itself.

---

### Decision · Program_Chairs · 2026-04-30

**Decision:**

Accept (regular)

**Comment:**

This is an emergency meta-review.

The paper maps out an interesting position in response to the emergence of LLM agent capabilities, which remove friction from interoperating between closed systems with public APIs.

Main area for improvement:
There is a general request from reviewers to substantiate some of the key assumptions with further evidence, which would strengthen the position advocated in the paper.

Recommendation and rationale:
Despite an assertion that the position is already widely adopted, this paper brings a clear framing to the issues that are emerging.  This appears to be a timely call to action founded on a literate and clearly reasoned analysis, which suggests that it would be a worthwhile contribution to the position paper track.